# Laser Triangulation Sensors Performance in Scanning Different Materials and Finishes

**DOI:** 10.3390/s24082410

**Published:** 2024-04-10

**Authors:** Victor Meana, Pablo Zapico, Eduardo Cuesta, Sara Giganto, Susana Martinez-Pellitero

**Affiliations:** 1Department of Construction and Manufacturing Engineering, University of Oviedo, Campus of Gijon, 33203 Gijon, Spain; zapicopablo@uniovi.es (P.Z.); ecuesta@uniovi.es (E.C.); 2Area of Manufacturing Engineering, University of Leon—Universidad de León, Campus de Vegazana, 24071 Leon, Spain; sgigf@unileon.es (S.G.); susana.martinez@unileon.es (S.M.-P.)

**Keywords:** non-contact metrology, laser scanning, outliers, point cloud filtering, reference spheres

## Abstract

The variety of equipment implementing laser triangulation technology for 3D scanning makes it difficult to analyse their performance, comparability, and traceability. In this study, three laser triangulation sensors arranged in different configurations are analysed using high precision spheres made of different materials and surface finishes. Three types of reference parameters were used: diameter, form error, and standard deviation of the point cloud. The experimentation was based on studying the quality of the point clouds generated by the three sensors, which enabled us to find and quantify an edge effect in the horizon of the scanned surface. A procedure to reach the optimal filtering conditions was proposed, and a chart of recommended usage of each sphere (material and finish) was created for the different types of sensors. This filter enables removal of both spurious points and those few points that spoil the form error, greatly improving the quality of the measurement.

## 1. Introduction

Industry increasingly requires the use of non-contact measurement systems to improve metrological control procedures, adopting new working models and adapting the procedures to new inspection technologies [1]. The rise of laser triangulation technology, which has been under development for decades [2], is increasingly used owing to its fast and high-precision scanning capacity. These features enable the widespread use of the technology in metrological tasks in industrial environments, even over and above reverse engineering tasks. The traceability of measurements is currently one of the main problems to be solved in this field, even despite the emergence of standards aimed at verification, coordinate measurement [3], and significant improvements in the accuracy of the most recent sensor models. Therefore, when it is necessary to incorporate optical equipment that needs to be qualified, the use of suitable reference elements is relevant [4]. Regarding verification procedures, interim checking, external/internal calibration procedures, etc.; the use of reference elements—usually precision spheres—is common [5,6,7]. These spheres enable the laser sensor to be calibrated in a similar way to the touch trigger probe, such as those mounted on coordinate measuring machines (CMM). The spheres are also used in alignment extension (increasing the working field) and as registration targets. However, the comparative analysis of different precision spheres, including those used to qualify laser sensors, has not yet been conducted. Specifically, in the present study, we compared the performance of three devices with laser triangulation technology by measuring reference spheres, while also considering the influence of different materials and their finishes.

The evolution of non-contact measurement equipment using laser technology is overcoming the disadvantages arising from the transition from laboratory use [8] to industrial applications. However, systematic errors in measurements occur [9,10], and some factors influence the suitability of the use of the equipment in metrology [11]. Comparative analysis of equipment must take the scanning strategy into consideration [12,13,14], which is, in essence, completely different between fixed and portable equipment. On the one hand, it is important to determine the direction of the scans as well as the number of scans to be carried out. On the other hand, some factors such as depth of field (DOF), field of view (FOV), and self-occlusion can limit the optical resolution of the process [1]. Other important considerations include the influence of the item to be measured in terms of the material it is made of, the surface finish, colour [15,16,17], and even the geometric characteristics [18,19].

Such is the case of the comprehensive research carried out at the National Physical Laboratory (UK), where they tested 3D optical scanner changes in measured dimensions due to artefact illumination, instrument temperature, scanner orientation, artefact surface colour, material and finish, and artefact position within a measurement volume [20,21,22].

In any case, the application of 3D laser technology to metrology is conditioned by the number of points acquired and their quality, and the influence of the aforementioned factors has therefore been studied for different fields of application and from different perspectives in order to improve scanning results [20,21,22,23,24,25,26,27,28].

In the present research, precision reference spheres made from a variety of materials and with different finishes were used to study artifacts with different optical properties. The study was carried out with three different triangulation sensors: one device mounted on a coordinate measuring machine and two portable devices, both mounted on coordinate measuring arms, with different measurement precisions. The first sensor is fully automatic in operation (trajectories and parameters can be automated) while the other two are portable devices and scanning is fully manual. In addition, detailed study of the different types of filters required to improve the quality of the point clouds—and therefore increase the precision of each type of equipment (and reference sphere)—was also conducted.

The performance of the three 3D laser sensors was analysed by establishing the relationship between the result of the measurements carried out on different spheres and the established reference obtained by contact measurement at the CMM of these spheres. Metrological traceability is therefore a factor to be analysed. In this respect, Carmignato et al. analysed the characteristics of numerous dimensional standards and classified them into different categories according to the type of measurement (linear, shape and surface texture, complex geometry, and angle) [29].

However, the relationship between optical measuring equipment and dimensional standards is a field of work that has not yet been studied in detail, mainly due to the rapid evolution of the associated technology. The research reported here provides relevant information in this respect. On the one hand, the geometrical shape used to determine the dimensional patterns was the sphere, as inspection of this element guarantees the traceability of comparisons [30]. On the other hand, different materials and finishes were used, as traceability-related problems derived from surface topography, and which arise when calibrating optical instruments are well known [31]. Thus, once the spheres have been measured using the coordinate measuring machine (CMM) based on two parameters (diameter and form error), the necessary reference data are available to evaluate the suitability and best technological solution (optical equipment) that adapts to each sphere and, therefore, to each material/surface finish.

Finally, this study also includes an in-depth analysis of the treatment (filtering process) of the point clouds captured with each of the laser devices. This is imperative given that it is almost always necessary to eliminate points classified as “noise”. The noise originates either from points that are not part of the surface scanned (points on the tooling, supports, etc.) or from spurious points, generated by reflections or by scanning the horizon or edge of the scanned surface. Based on this in-depth analysis of the filters applied to eliminate or minimise the noise, it is suggested which filtering level should be used so that the sphere detected with the sensor is as similar as possible to the reference sphere. Thus, the material and its finish are correlated with the optimum filter configuration for the sensor. The study findings also enable identification of the reference sphere that will best allow calibration of a sensor with similar characteristics to any of those analysed.

Using the data obtained and by applying a filter based on the standard deviation of the point cloud, it is possible to determine the optimum filtering level for each sensor depending on the sphere measured. Thus, the material and surface finish that characterises the standard (sphere) can be identified for calibration of laser triangulation devices with similar technical characteristics to any of those analysed in this study.

## 2. Materials and Methods

Reference standards are essential elements in contact coordinate metrology (e.g., coordinate measuring machines, CMMs), as elements that are used to periodically qualify sensors, interim checking, instant verification, and certification of measurements. However, even in the case of 3D sensors, reference standards are required for qualification and calibration and to ensure the traceability of the measurements [32]. The most commonly used elements are spheres, as they are canonical geometric forms whose diameter and form error can be precisely and simply specified, both mathematically and metrologically. In fact, measurement software already incorporates routines and optimised procedures for measuring spheres. The position of the centres, the diameters of the spheres, and the form error are metrologically stable and consistent characteristics. The use of this geometry as a reference feature is also widely used in other applications—such as industrial photogrammetry, total station, target-based triangulation, and laser tracking [33]—or in the calibration of computed tomography (CT) devices [34].

In this research, 6 spheres are used with similar diameter ranges, Ø20 to Ø25 mm (Table 1), made from three different types of material (metal, ceramic, and polymer), and with two types of finish (glossy and matte). The characteristics of each type are summarized in Table 1. All spheres are commercial precision spheres (quality grades G10 to G100 according to ISO 3290-1 [35]). In terms of the materials, the spheres were a stainless steel AISI 316L sphere (ref. Sb-1), a polymer-coated AISI 440C steel sphere (ref. Co-1), a tungsten carbide sphere (ref. TC-1), and three different ceramic spheres (Ce-i). One of the ceramic spheres was made from pure zirconium oxide (ZrO_2_, ref. Ce-1), one from pure alumina (Al_2_O_3_, ref. Ce-3), and the other from a mixture of alumina and zirconium oxide (10% ZrO_2_ and 90% Al_2_O_3_, ref. Ce-2), referred to as “Aluzir” or ZTA (Zirconia Toughened Alumina [36]). Half of the spheres (stainless steel, polymer-coated steel, and ZTA) had a matte finish, while the other three (tungsten carbide, ZrO_2_, and Al_2_O_3_) had a glossy finish (Figure 1).

On the other hand, a coordinate measuring machine (CMM), model DEA Global Image 091508 (X = 900 mm, Y = 1500 mm, Z = 800 mm) was used as the reference metrological equipment. This device was used to carry out high-precision contact measurements, enabling assessment of the metrological quality of the laser triangulation sensors. For this study, the machine was equipped with a Renishaw PH10MQ indexing head with an SP25M probe, to which a 30 mm long ceramic stylus with a Ø4 mm ruby sphere end was attached. The machine control uses the PC DMIS 2018 R2 Computer Aided Inspection software. This control enables automated programming and execution of measurement routines, which allowed a uniform and dense distribution on the spheres, confirming the calibrated values of diameter and form error of the spheres. Regarding the metrological performance of this machine, the maximum permissible error in the length indication fits the equation E_0,MPE_ = 2.2 + 3 L/1000 µm (L in mm), while the maximum radial error in repeatability is R_0,MPL_ = 2.2 µm.

As for the laser triangulation sensors used for non-contact measurements of the spheres, three sensors were used in two different arrangements (Table 2), in terms of their portable capacity and potential automation. One of them is a sensor coupled to the indexable head of the CMM, with automatable parameters, orientations, and trajectories. Specifically, it is the sensor model HP-L-10.6 from Hexagon Metrology (LS-CMM). The other two were fully manually operated portable sensors, in both cases attached to the wrists of two coordinate measuring arms (AACMM or CMA). One of them is the RS6 Laser Scanner (LS-ARM-1), mounted on an Absolute Arm 7-axis. The scanner sensor specifications are a data rate of 1.2 million points/sec and a probing form error of 26 µm, certified according to ISO 10360-8:2013 [3]. The second manually operated laser sensor, the Romer Sigma R-SCAN (LS-ARM-2), was mounted on a Romer Sigma Portable Measuring Arm. This second portable sensor is from a generation previous to the RS6, and therefore of lower accuracy, as its probing error is 44 µm and with a maximum point acquisition rate of 19,200 points/sec. The technical specifications of the three available laser sensors are compared in Table 2, which shows the technical capacities and difference in accuracy of each.

## 3. Experimentation

The methodology followed in the experimentation (Figure 2) was divided into five sequential phases, each corresponding to the following activities:(a)Contact measurement of the spheres and establishment of the main reference parameters (diameter and form error) in order to produce the reference values (both dimensional and geometrical) with maximum accuracy.(b)Non-contact measurement of the spheres. Each of the six spheres were measured with the three laser triangulation devices to obtain the point clouds. The most appropriate parameters were used in each case to reliably obtain high-quality point clouds.(c)Cleaning, trimming, and filtering of the point clouds. The inspection software was used to trim the point clouds, and the spurious points were cleaned by applying the corresponding filters.(d)Determination of the analysed parameters (number of points captured, diameter, form error, and standard deviation). Once the appropriate filter was applied, the spheres were fitted with the remaining points, obtaining the values of the parameters to be compared with the reference ones (form error and diameter), as well as the parameters used to characterise the behaviour of the equipment (number of points and standard deviation).(e)Analysis of results. The results obtained were analysed in relation to the equipment and spheres used, as well as the filters applied.

Contact measurement of the precision spheres was carried out with the DEA Global Image CMM according to ISO 10360-5:2020 [37] with the upper hemisphere of each sphere as the reference area. In this area, 50 points distributed into nine meridians were recorded. The procedure was repeated 10 times for each sphere; average values were obtained and checked to ensure that the differences between the 10 measurements were always less than 0.002 mm (below the maximum scanning error of the CMM). This procedure enabled determination of the diameter and the reference form error. The latter is defined as the radial difference between the outermost and innermost points of the best-fit sphere. The contact measurement strategy followed for the different spheres, and an image of the stylus during one of the measurement processes are shown in Figure 3.

The six spheres were then scanned with the three laser triangulation devices. In the case of the automatic equipment (Sensor HP-L-10-6, LS-CMM) and in order to obtain adequate coverage of the spheres, the points were captured from five different orientations: four with the sensor tilted at 45° and one with the sensor oriented vertically downward. One of them is shown in Figure 4a, and spatial orientation axes are shown in Figure 4b. The appropriate parameters were configured using the PC-DMIS control software to obtain the best capture on all spheres. This configuration corresponds to a normal gain, an average or standard point density of 16.8 points/mm, and a line width of 123 mm, while the incidence angle filter was set at 75°.

The above experimentation methodology was also applied to two other laser triangulation devices, in this case portable (Figure 5a,b), where human interaction has a greater influence. The operator is required to have the necessary experience and skill to achieve adequate coverage of the geometry to be captured with the minimum possible exposure and time while also minimising the dispersion of values. In both cases, the capture of the point clouds was performed 10 times and by two different operators. Before starting the data capture, the sensors were qualified using procedures recommended by the manufacturer, to maximise accuracy.

The raw point clouds captured with the sensors (Figure 6a) were treated with Geomagic Control X inspection software to first remove all the information generated that did not belong to the spheres being scanned. The cleaning was continued by removing points that were below the established reference plane (with a spherical coverage angle φ=220° to ensure that at least the hemisphere points are always captured) in order to homogenise the data acquired with the three sensors (Figure 6b).

Once the spheres were cleaned of noise (fragments and non-sphere elements), a mathematical sphere was fitted to the scanned point clouds, in this case using the least squares method. This “best fit” mathematical sphere yields the diameter and the position of the centre of the sphere. The quality of each individual point captured can be determined from both values, and distances or bands of distance from the mean value (diameter of the best-fit sphere) can be established. For these ranges, the standard deviation parameter (σ) and its multiples (k-factors) were used. Considering that the distribution of the data obtained follows a normal distribution (Figure 7), we can use a multiple k of the standard deviation (σ) as a filter that rejects or eliminates the points furthest away from the mean diameter. Thus, for example, a filter of ±3σ (6σ amplitude) would leave the cloud intact (eliminating only 0.27% of the furthest points), while a filter of ±1σ would leave the cloud with only the points in the central zone, eliminating 31.73% of the points furthest from the diameter (or, in other words, leaving 68.26% of the closest points).

Figure 8 shows the zenith views of the point clouds obtained with the LS-CMM automatic sensor versus one of the portable sensors (LS-ARM-2). The first view (Figure 8a) shows the banding patterns generated by the horizon effect, which causes the accuracy of the points captured in these areas to decrease drastically as the laser beam is almost tangential to the surface. The second view (Figure 8b) shows the completely random line patterns generated by the different filters used in the manual scanning. This effect of curvature on the measurement quality of sensors similar to those studied has been reported by other authors [38,39] in the measurement of surfaces with a large variation in curvature, such as gear profiles.

To estimate the uncertainty associated with the measurement results obtained in this study, the procedure recommended by the Guide to the Expression of Uncertainty in Measurement (GUM) can be applied. This procedure involves the quadratic propagation of uncertainties from the various influencing factors. These factors include the standard uncertainty of calibration of the spheres, $u_{Sph}$. Despite some spheres having calibration certificates, we consider it an overestimation that the expanded uncertainty assignable to each sphere is the maximum allowed by the used CMM (R_0,MPL_). On the other hand, the standard uncertainty associated with each sensor is obtained from the information provided by each manufacturer (probing error), $u_{SenMan}$, considering a coverage factor k = 2. Thus, the estimation of uncertainty for each sensor (subscript i), $u_{i}$, is as shown in Equation (1). The contribution from temperature variations is disregarded, as all measurements were conducted in a metrology laboratory with controlled temperatures within the range 20±1 °C.
(1)$$u_{i}=\sqrt{u_{Sph}^{2}+u_{{SenMan}_{i}}^{2}},being\;i=1\;to\;3$$

Thus, Table 3 presents the estimation of uncertainty for the analyzed sensors.

## 4. Results and Discussion

Once the methodology was established and the parameters that will allow the analysis of the point clouds of each sensor were defined, this section presents the results obtained. The reference values obtained with the contact CMM are shown in Table 4. The geometrical quality of all the spheres is consistent with the nominal accuracy and with the measurement capacity of the CMM. It is also consistent with the manufacturing process and finishes of the spheres, with measurements of the sandblasted and polymer-coated spheres being less accurate, although the error in the values did not exceed 5 µm in any case.

Based on these data and bearing in mind that the main objective of the research was to analyse and evaluate the performance of the three laser triangulation sensors used to scan the different materials and finishes, the quality of the point clouds captured from the six reference spheres was then compared. Thus, once the cloud was trimmed, leaving the points between the reference plan of each sphere and the pole, and the spurious points removed, each point cloud captured with each of the three sensors and for each sphere was examined (Figure 9). Regarding the standard deviation of the best-fit clouds, significant differences were observed between sensors, both quantitatively (point density) and qualitatively (cloud standard deviation). As the diameter of the spheres was different, the bar graphs in Figure 9 used the density of the point cloud (expressed in points/mm^2^) instead of the number of points.

On the one hand, it is significant that neither the lower-performing portable sensor LS-ARM-2 or the automatic sensor LS-CMM were capable of capturing information from the tungsten carbide sphere (TC-1), which had the shiniest finish.

In addition, the LS-ARM-2 sensor yielded very different results depending on the sphere scanned, with contrasting values for two of the spheres with a matte finish. The dot density for the sphere painted with polymeric material (Co-1), 62 points/mm^2^, was much lower than obtained for the ceramic sphere (Ce-2), 1286 points/mm^2^.

Regarding the point density, more homogeneous measurements were obtained for all of the spheres scanned using the most modern sensors (the automatic LS-CMM and the portable LS-ARM-1). The LS-ARM-2 sensor is extremely sensitive to the strategy chosen for manual scanning, while the other two sensors incorporate “ad-hoc” software filtering, which avoids duplication of points with coordinates that are very close together. In fact, these sensors automatically eliminate points whose coordinates are very close to others that have already been recorded, allowing for less dense capture without loss of precision.

The range of the point cloud density for the six spheres scanned with the different sensors can be observed in Table 5. Once again, the more modern LS-CMM and LS-ARM-1 sensors performed best in this respect. The differences between the minimum and maximum values of the point density generated by these sensors for the different spheres were 38 points/mm^2^ and 100 points/mm^2^ respectively. Of these, the RS6 sensor yielded the highest point density, an average of 485 points/mm^2^ for all spheres.

In regard to the scan quality, the standard deviation σ of the cloud relative to the best-fit sphere provides useful information (Table 5). The LS-ARM-1 sensor (RS6) is extremely constant (standard deviation between 10 and 20 µm), followed by the automatic LS-CMM sensor (HP-L-10-6). The LS-ARM-2 sensor (R-SCAN) again yielded the most variable results depending on the spheres, both for the absolute values and the dispersion, with values ranging from 45 µm to 78 µm. The manual scanning greatly affected this model, which does not incorporate default noise reduction routines for the duplication (proximity of coordinates) of scanned points as the LS-ARM-1 sensor does.

The parameters provide a good idea of the influence of the capacities and performance of the equipment for use with different materials and surface finishes.

### Analysis of the Influence of the Filter

There is a clear relationship between the standard deviation (σ) of the point clouds and the form error yielded by the sensor. An ideal sensor of maximum precision would generate a point cloud whose form error of the sphere would be equal to the form error obtained by contact. By geometric form error, we mean the difference between the outermost point and the innermost point of each point cloud (“Dmax-Dmin”).

We therefore studied the influence of the application of a “Sigma” type filter or k·σ filter (k being the multiple of the standard deviation) on the form error. The aim of this study is to determinate the capacity of this type of filter to eliminate the furthest points, but without distorting the measurement, improving the metrological performance of the sensors for each sphere. The values obtained for the three sensors and the six reference spheres were plotted in a graph (Figure 10), in which the standard deviation values correspond to the raw cloud, removing obvious outliers (equivalent to applying a k > 6 filter).

While the laser sensor LS-ARM-1 (RS6) provides the lowest values of the three devices and use of the filter had comparatively less influence (values of around 90 µm for any k·σ filter range always below 61 µm), the form error obtained with the automatic LS-CMM (HP-L-10.6) sensor was clearly dependent on and proportional to the k·σ filter. In this case, the least suitable sphere was the zirconium oxide sphere (Ce-1 ceramic), which produced values of 510 µm for scarcely filtered data (k = 6) and 184 µm for k = 2 (±2σ or 4.2% of the discarded points). On the other hand, this sensor yielded the best results for the sphere with the polymeric material coating (Co-1), and the values were less dependent on the filter applied (linear response and lower slope). As for the values obtained with the third sensor, the LS-ARM-2 (R-SCAN), the heterogeneity of the results with each sphere and with the application of the different filters is evident. The high level of variability can be explained by the age of the sensor and its poorer performance. The form error yielded by this sensor with the sphere with which the best results were obtained (Ce-2 ceramic) was greater than the form errors yielded by the other two sensors with this sphere (for any k with the RS6 sensor and for k > 2 with the HP-L-10.6 sensor).

By comparing the diameter measurements obtained with the sensors and the reference measurements obtained with the contact CMM (Figure 11a) and, similarly, comparing the form errors yielded by the three sensors and the contact CMM (Figure 11b), we can analyse the quality of the reconstructions of the spheres as a function of the filters ±k·σ applied (all with a coverage angle φ=220°, Figure 5b).

Thus, in the study of the differences in the diameters evaluated (Figure 11a), we can see that, for the spheres measured with the manual sensors (LS-ARM-1 and LS-ARM-2), the two laser sensors generate point clouds whose diameter deviations relative to the reference point clouds are independent of the filter level (k·σ). Only the automatic sensor LS-CMM was somewhat influenced by the type of filter, although only strong filters, k = 2 or k = 1, produced substantial variations in the measured diameter.

The values of diametric deviations (Figure 11a) varied greatly from one sensor to another and with the different spheres. The LS-CMM sensor was very sensitive to the type of material and its finish, as it produced better measurements of the diameters of the matte-finished spheres, such as sandblasted (Sb-1) or coated (Co-1) spheres. At the other extreme are the glossier spheres (ceramic material such as Ce-3 and Ce-2) and the glossiest TC-1 sphere, which cannot even be measured with this sensor. With the LS-ARM-1 sensor, all spheres can be measured with good repeatability (independently of the filter), even the glossiest sphere (TC-1). In general, the differences relative to the reference measurements (CMM) were significantly lower than those produced by the LS-CMM automatic laser. Focusing on the manual LS-ARM-2 sensor, the differences between spheres were even greater, varying between the sandblasted Sb-1 sphere (+100 µm) and the Co-1 sphere (−400 µm). The scarce influence that the filtering had on the diameter value measured by this sensor is striking. Thus, from the point of view of diameter measurement, there is no need to remove many points from the clouds, and they can be left almost intact by applying very weak filters (e.g., k = 6, ±6σ), which are sufficient to eliminate spurious points and minimise the “horizon effect” or “edge effect”.

Focusing on the deviations of the form error FE (Figure 11b), very low values of this parameter were obtained with the CMM for the spheres (FE_CMM_ < 5 µm), especially in comparison with the form errors of the adjusted spheres of each one of the sensors (FE_f_). Therefore, the graphs constructed for FE_f_-FE_CMM_ are similar, with the values almost the same as those represented in Figure 10. However, the performance of the sensors, in terms of the form error, is included in Figure 11b, modifying and adapting the scale of each of the graphs of each sensor to observe the influence of the filter k·σ value in greater detail (mainly in the LS-ARM-1 sensor).

## 5. Conclusions

This paper reports a study of the performance of three laser triangulation sensors for scanning different precision spheres: an automatic sensor coupled to a fixed CMM; and two manually operated, fully portable sensors, coupled to two AACMMs of different generations. These sensors, with different resolutions and similar fields of application, were used to scan precision spheres made of six different materials and with two types of surface finishes. Precision spheres were chosen as reference elements for the study because they are widely used and commercially available; they are commonly used as verification, adjustment, and/or calibration standards. In addition, they are easy to characterise from a mathematical and metrological point of view.

To assess the possible influence of the finishes of the spheres, half of the samples had a glossy finish while the other half had a matte finish. The study is based on reference information for each sphere, obtained from high-precision contact measurements using the CMM. The extent to which the reconstruction of the spherical geometries from the point clouds obtained with the sensors is close to the first ones in geometric and dimensional quality was then analysed. Finally, the work was completed by studying the influence of using a standard filter, the “σ” filter, or “k·σ” filter, to eliminate residues based on the statistical distribution (which tends to a normal distribution) of the points generated. Under these premises, the following conclusions were reached:

Analysis of the capacity of the sensors to capture points on the different spheres showed that the LS-ARM-1 sensor (Romer RS6) produced the best results, both quantitatively (higher density of points captured) and qualitatively (homogeneity between the different spheres). This was also the only sensor capable of capturing points on polished, glossy spheres (TC-1). The less accurate LS-ARM-2 sensor (Romer R-SCAN) produced the worst results, producing disperse values depending on the type of sphere (the best results were obtained with ceramic material and matte finish, and worst results with any glossy finish), even not capturing points on the glossiest sphere, the polished TC-1 sphere.

The qualities of the point clouds, which are also determined, among other things, by the standard deviation, also differed between the three sensors when filtering was not applied (k >> 6). The LS-ARM-1 sensor again produced the best results (10 µm for the Co-1 sphere), ahead of the LS-CMM (18 um for the Sb-1 sphere) and the LS-ARM-2 (45 µm for the Ce-2 sphere of Aluzir). The fact that the best value was obtained with the Co-1/LS-ARM-1 pairing is reasonable if we consider the capacity and performance of this new generation sensor, and that these coated spheres are usually supplied for calibration and qualification of laser sensors. The glossy ceramic spheres (Ce-3 made from Al_2_O_3_ and Ce-1 made from ZrO_2_) yielded the poorest results with almost all sensors, with a value of up to 78 µm in the standard deviation of the cloud with the R-SCAN sensor.

On the other hand, considering the geometrical quality of the spheres, understood as the deviation from the form error of the reconstructed spheres (Figure 9), the filtering level has a strong influence. However, this influence was quantitatively smaller with the LS-ARM-1 sensor than with the other two sensors. The LS-ARM-2 proved to be the worst performing of the three devices, and it was more susceptible to including noise in the point clouds. Detailed analysis (Figure 11b) of the data produced by the LS-ARM-1 sensor indicated that strong filters (k = 3 or even k = 2) were required before the effects of the point removal were observed in the form error value. For the other two sensors, the deviations of the form errors were much higher than for the LS-ARM-1 sensor, reaching the order of one millimetre in the case of the R-SCAN with the Ce-1 (ZrO_2_) sphere, so that use of this sphere with this sensor should be ruled out.

In the sensor with automatable trajectories (LS-CMM), the linearity of the performance regarding the form error is very noticeable when filters are applied (Figure 11b). The fact that the trajectories can be automated and capture the different spheres in the same areas undoubtedly generated the high linear effect when applying filters. The slopes of the straight lines of the different spheres were higher for glossy spheres than for matte spheres, so the matte spheres are suitable for qualifying this type of sensor (Table 6).

In summary, the best spheres for calibrating or qualifying each of the sensors are shown in Table 6. Although the sigma filtering recommended for working with diameters is sufficient (k = 6), in the case of form error of the spheres, use of a strong filter (range k = 2) is advisable, as this substantially improves the quality of all the point clouds captured by laser triangulation sensors.

A study regarding the best type of filter, which is always necessary to apply with this type of sensor, was carried out. This filter enabled removal of both spurious points and points that spoil the form error, greatly improving the measurement that these sensors can obtain from point clouds. In particular, as the filter that yields the best diameter and form error measurements is considered the best, we can conclude that the sandblasted stainless-steel sphere (Sb-1) is quite suitable for the automatable sensor (LS-CMM) with application of a k = 2 filter. This sandblasted sphere would also provide acceptable results for the LS-ARM-2 sensor together with the Co-1 coated sphere. For the LS-ARM-2 sensor, the best results were obtained with the Aluzir sphere (Ce-2), also with a strong filter (k = 2).

## Figures and Tables

**Figure 1 sensors-24-02410-f001:**
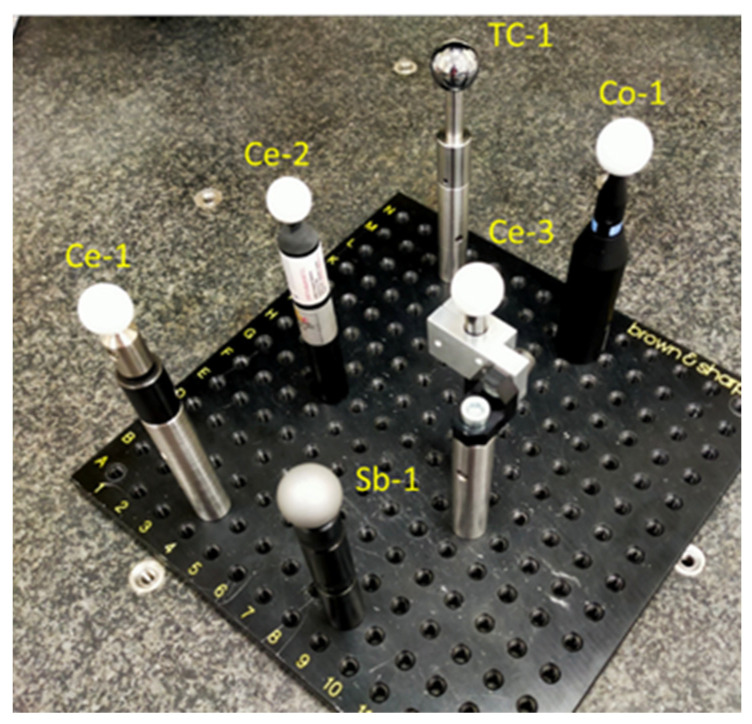
Spheres used in the study.

**Figure 2 sensors-24-02410-f002:**
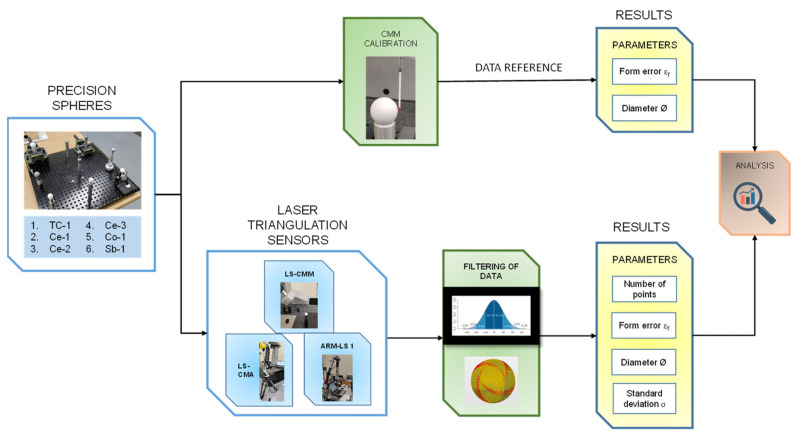
Experimental methodology.

**Figure 3 sensors-24-02410-f003:**
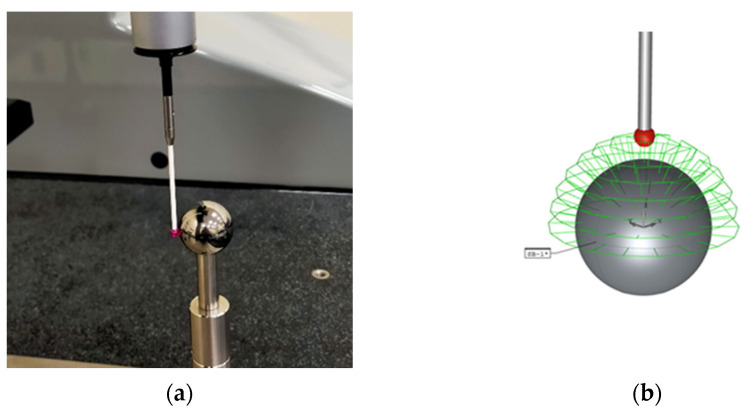
(**a**) Measurement made with the CMM on the Ø25 mm sphere. (**b**) Contact measurement strategy.

**Figure 4 sensors-24-02410-f004:**
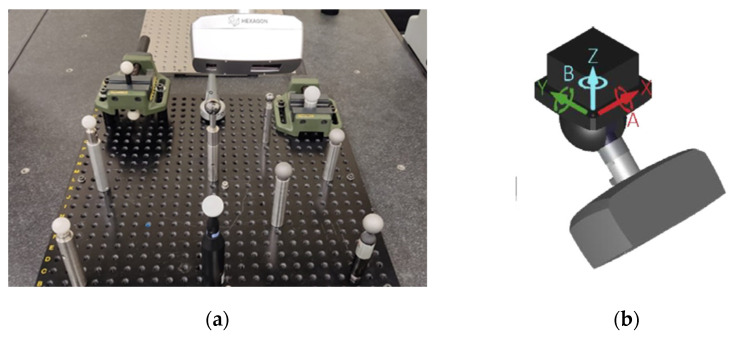
(**a**) 45° position of the HP-L-10.6 sensor (LS-CMM). (**b**) Sensor representation according to spatial orientation axes.

**Figure 5 sensors-24-02410-f005:**
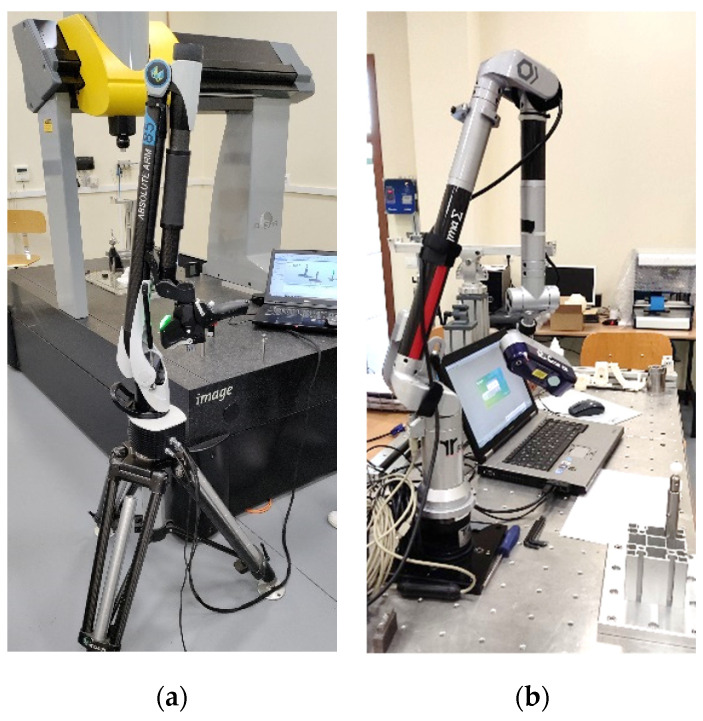
(**a**) RS6 Laser Scanner mounted on Absolute Arm 85 (LS-ARM-1). (**b**) R-SCAN mounted on articulated arm laser Romer Sigma (LS-ARM-2).

**Figure 6 sensors-24-02410-f006:**
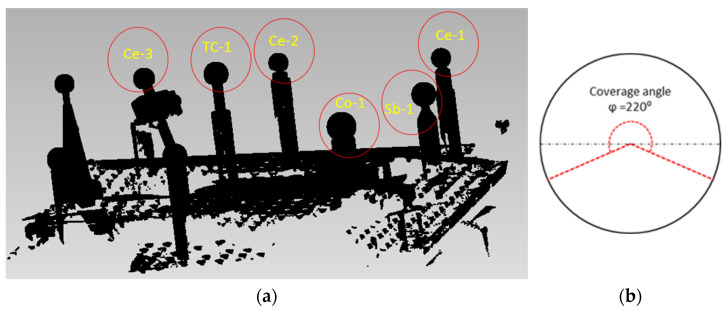
(**a**) Raw point cloud captured with LS-CMM sensor. (**b**) Spherical coverage angle.

**Figure 7 sensors-24-02410-f007:**
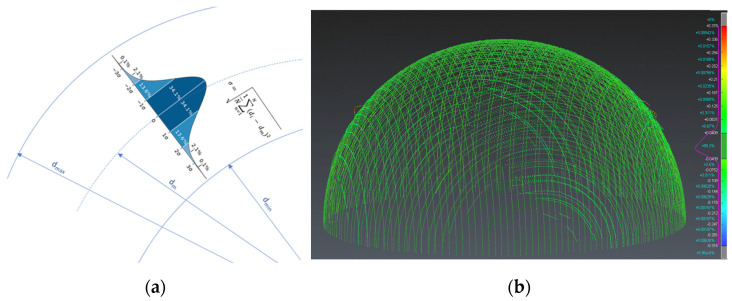
(**a**) Normal distribution of the point cloud on the best-fit sphere. (**b**) example of a point cloud and distribution analysis (highlighted in red, inside the circle, are the points removed with filter 2σ).

**Figure 8 sensors-24-02410-f008:**
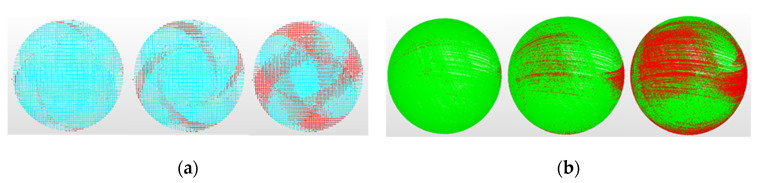
(**a**) Point clouds obtained with the LS-CMM sensor for k = 3, 2, and 1. (**b**) Point clouds obtained with the LS-ARM-2 sensor for k = 3, 2, and 1.

**Figure 9 sensors-24-02410-f009:**
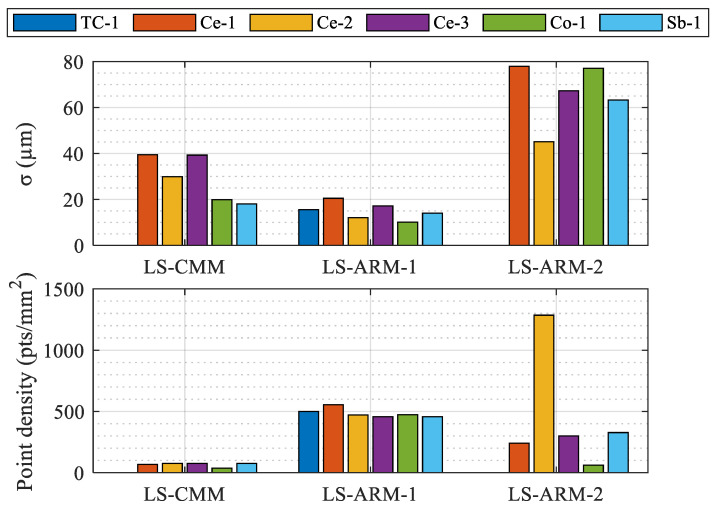
Quality of point clouds of spheres based on point density and standard deviation (σ).

**Figure 10 sensors-24-02410-f010:**
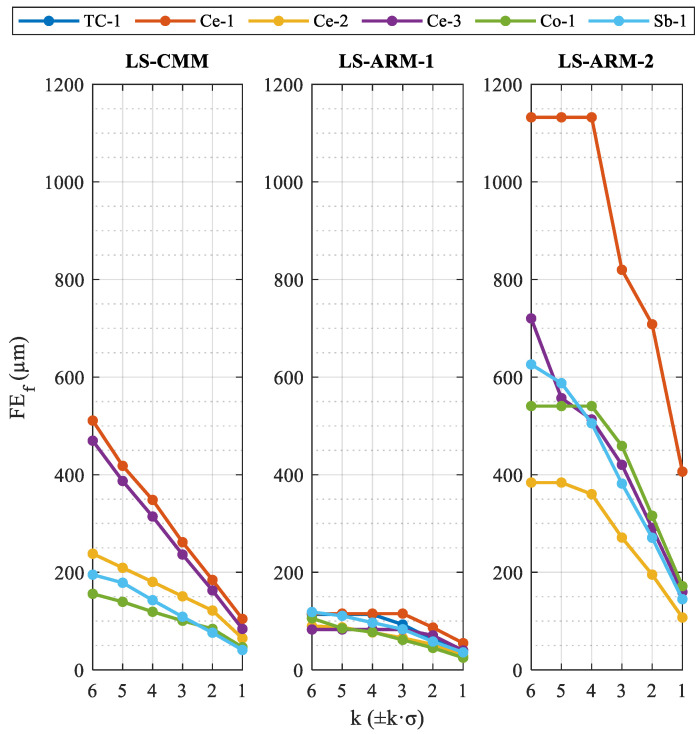
Influence of the filter based on multiples of standard deviation (k·σ) on the form error (FE) of the spheres, for each of the three laser sensors.

**Figure 11 sensors-24-02410-f011:**
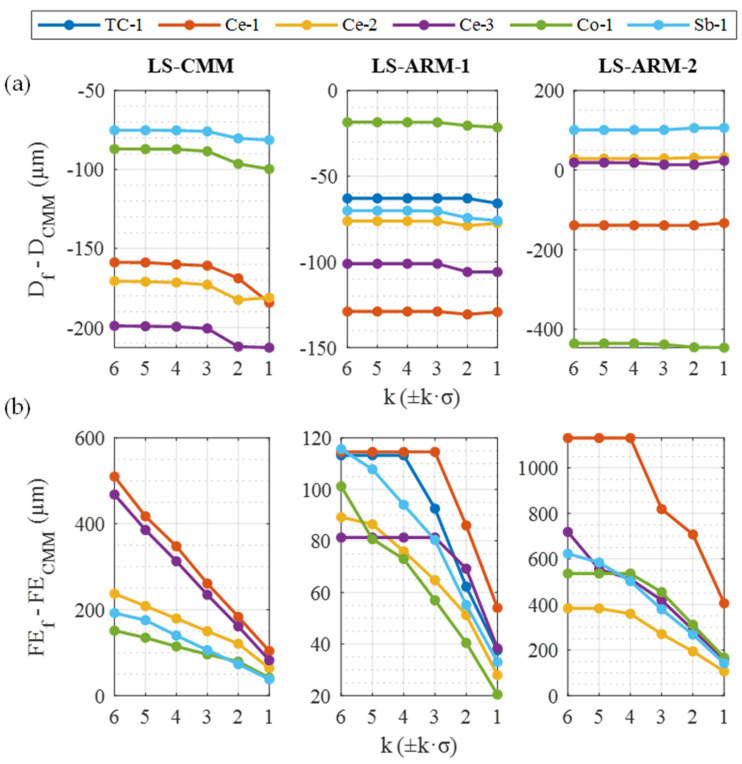
Sensor deviations, diameters (**a**) and form errors (**b**), from the reference values taken with the CMM as a function of the filter k·σ.

**Table 1 sensors-24-02410-t001:** Characteristics of the spheres used in the research.

Id.	Material	Finish	Nominal ∅D (mm)	Nominal Precision Grade (Deviation from Spherical Form, ISO 3290)
TC-1	Tungsten carbide	Glossy	25	G5 (0.13 µm)
Ce-1	Zirconium dioxide (ZrO_2_)	Glossy	20	G10 (0.25 µm)
Ce-2	ZTA (10% ZrO_2_, 90% Al_2_O_3_)	Matte	20	G40 (1 µm)
Ce-3	Alumina (Al_2_O_3_)	Glossy	22	G20 (0.5 µm)
Co-1	Polymer-coated steel (core of AISI 440C)	Matte	25	G200 (5 µm)
Sb-1	Stainless steel (AISI 316L)	Matte	25	G100 (2.5 µm)

**Table 2 sensors-24-02410-t002:** Laser line scanners used in the study.

Laser Triangulation Scanner Mounted on CMM (LS-CMM)	Laser Triangulation Scanner Mounted on Coordinate Measuring Arm (LS-ARM-1)	Articulated Arm Laser Romer Sigma R-SCAN (LS-ARM-2)
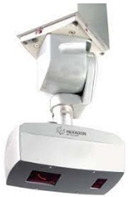	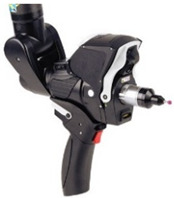	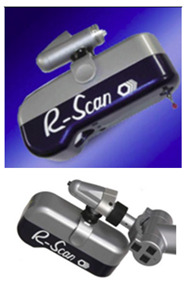
HP-L-10.6^®^ from Hexagon Metrology	RS6 Laser Scanner mounted on Absolute Arm 85 from Hexagon Metrology	R-SCAN from Romer Sigma
Data rate: 30,000 points/s	Data rate: 1,200,000 points/s	Data rate: 19,200 points/s
Stand-off and DOF: 170 ± 30 mm Min. point spacing: 0.030 mm	Stand-off and DOF: 165 ± 50 mm Min. point spacing: 0.027 mm	Scanning distance: 124–222 mm Min. point spacing: 0.100 mm
Line rate (max.): 53 Hz Laser line width: 24, 60 or 123 mm	Line rate (max): 300 Hz Laser line width (mid.): 150 mm	Line rate: 30 Hz Laser line width (max): 110 mm
Probing form error: 0.022 mm	Probing form error: 0.026 mm	Probing error: 0.044 mm

**Table 3 sensors-24-02410-t003:** Estimation of uncertainty budget for the sphere’s measurements (mm).

Contribution	Evaluation Type	Standard Uncertainty	LS-CMM (HP-L-10.6)	LS-ARM-1 (RS6)	LS-ARM-2 (R-SCAN)
$u_{Sph}$ Calibration gauge (Sphere)	Type B, stated in calibration (CMM)	$u_{Sph}$	0.0011	0.0011	0.0011
$u_{SenMan}$ Laser sensor standard uncertainty	Type B, stated by manufactured ^1^	$u_{SenMan}$	0.0110	0.0130	0.0220
Combined standard uncertainty, $u_{i}$:			0.0111	0.0130	0.0220

^1^ This evaluation was Type A for the manufacturers.

**Table 4 sensors-24-02410-t004:** Reference values (CMM contact measurements).

Identifier	Material	Finish	CMM Measurements	
			Diameter (mm)	Form Error (µm)
TC-1	Tungsten carbide (WC)	Glossy	24.9994	0.4
Ce-1	Zirconium dioxide (ZrO_2_)	Glossy	19.9995	0.8
Ce-2	Aluzir (ZTA)	Matte	20.0021	0.8
Ce-3	Alumina (Al_2_O_3_)	Glossy	22.0005	1.5
Co-1	Polymer-coated steel (AISI 440C)	Matte	25.4878	4.6
Sb-1	Stainless steel (ASIS 316L)	Matte	25.0100	2.6

**Table 5 sensors-24-02410-t005:** Quality of the raw point cloud (without filtering).

	LS-CMM (HP-L-10.6)		LS-ARM-1 (RS6)		LS-ARM-2 (R-SCAN)	
Sphere	σ (μm)	PD (points/mm^2^)	σ (μm)	PD (points/mm^2^)	σ (μm)	PD (points/mm^2^)
TC-1	-	-	16	500	-	-
Ce-1	39	67	20	556	78	242
Ce-2	30	75	12	471	45	1286
Ce-3	39	75	17	456	67	300
Co-1	20	37	10	473	77	62
Sb-1	18	75	14	457	63	327

**Table 6 sensors-24-02410-t006:** Recommended usability for sensor calibration and qualification.

Sphere	LS-CMM (HP-L-10.6)	LS-ARM-1 (RS6)	LS-ARM-2 (R-SCAN)
TC-1 (WC, glossy)	X	√	X
Ce-1 (ZrO_2_, glossy)	X	X	X
Ce-2 (ZTA, matte)	√	√√	√√
Ce-3 (Al_2_O_3_, glossy)	X	√	X
Co-1 (white coated, matte)	√√	√√√	√
Sb-1 (AISI 316, matte)	√√	√√	√

## Data Availability

The raw data supporting the conclusions of this article will be made available by the authors on request.

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
