# Peer review of "Laser Triangulation Sensors Performance in Scanning Different Materials and Finishes"

_sensors, 2024, doi:10.3390/s24082410_

Round 1

Reviewer 1 Report

Comments and Suggestions for Authors

The article delves into an experimental investigation of laser triangulation technology, focusing on its metrological performance and traceability, which remain significant challenges. This study conducts experimental research using three distinct laser triangulation sensors across a range of spheres to thoroughly evaluate and characterize the scanning performance.

 In my view, while the subject matter is intriguing, it feels incomplete without the development of an uncertainty budget for the laser triangulation technology. Such a budget is crucial as it would systematically analyze all potential error sources relevant to this technology, providing a more comprehensive understanding of its limitations and capabilities.

I find the topic to be highly relevant in the field, especially considering the growing adoption of this measurement principle in the industry for the realization of in-process metrology. However, I don’t feel that it addresses a specific gap in the current state of the art.

The experimental research is relevant compared to other published material. However, I would say that it is crucial to analyze all the potential error sources that affect the measurement performance of this technology to provide a more comprehensive understanding of its limitations and capabilities.

The article presents a specific experimental study of three laser triangulation models. This is the main reason why I suggest researching the potential error sources that affect the measurement uncertainty of this technology because this approach would help to address specific improvements beyond the current state of the art.

The conclusions align with the experimental research conducted, yet they are not broadly applicable, which limits the scope of this study's application.

The references are appropriate up to some point. I would suggest adding the research work performed by NPL some years ago where researchers such as Martin Dury and Andrew Lewis were involved.

Comments on the Quality of English Language

Minor revision is needed.

Author Response

Please find enclosed the response file

Reviewer 2 Report

Comments and Suggestions for Authors

The authors present an interesting comparison study regarding the measurement capabilities of different laser triangulations sensors applied on different object materials. The main question is the different measurement quality when comparing three different state-of-the-art laser triangulation sensors. In addition, the influence of the material/surface finish on the measurement quality is discussed.

Studies that address the influence of different materials and surface properties are typically on a more qualitative level. Here, the measurement quality is compared for multiple different surface/material types which is a substantial distribution providing an experimental, well-documented dataset. The study addresses a gap in the field, because studies with a clear description of the experimental conditions and the reference are missing.

I appreciate the sensor characterization study while I must admit that the value of the study is no contribution of a fundamental scientific finding but a practical-oriented study that clarifies the measurement capabilities of existing sensors. However, it seems to be exactly within the scope of sensors, so I can recommend it for publication.

Minor issues: I recommend to enrich the discussion of the surface topography details (of the different measurement objects) and the corresponding light scattering behaviour, to add a more quantiative-based explanation of some of the measurement results.

I also would give the hint to address the error type that has been discussed in literature that the surface tilt and curvature respectively, has. Are the results in agreement with the theory that has been presented by others?

The references are appropriate, but they could be enhanced by those works which addressed similar error types of laser triangulation, in particular: https://www.mdpi.com/1424-8220/21/3/937 https://ieeexplore.ieee.org/document/9584898

Some formatting issues remain, such as writing mm^2 and \varphi, which should be double-checked.

Round 2

Reviewer 1 Report

Comments and Suggestions for Authors

I would suggest including a detailed uncertainty budget analysis considering the results obtained on this experimental research to let the reader know about the uncertainty contributors and their weight when using laser triangulation technology.

Comments on the Quality of English Language

-

Author Response

Please, find enclosed the response file
